# Modeling the Groundwater Dynamics of the Celaya Valley Aquifer

Ana B. Rubio-Arellano [1],[†] , Jose A. Ramos-Leal [1],[†] and Víctor M. Vázquez-Báez [2],*,[†] and José I. Rodriguez Mora [2],[†]

1 Instituto Potosino de Investigación Científica y Tecnológica, División de Geociencias Aplicadas, Camino a la Presa de San José 2055, Lomas 4ta Secc, San Luis Potosí 78216, Mexico
2 Facultad de Ingeniería, Benemérita Universidad Autónoma de Puebla, Puebla 72570, Mexico
* Correspondence: manuel.vazquez@correo.buap.mx
† These authors contributed equally to this work.

**Abstract:** We propose a hydrodynamic simulation model to understand the piezometric operation of the Celaya Valley aquifer. The aquifer is located at the east of Guanajuato State in México. Our proposed model reproduces, under transitory conditions, the performance of the aquifer during the dry season from year 2015 to 2019 and the rainy season from year 2010 to 2015. The simulation was projected for the two seasons up to eleven years ahead and considering only three simulation scenarios (one should bear in mind that there are a number of different and interesting scenarios): trend, pump reduction and pump increase. In general terms, the model accurately reproduces the natural conditions of the aquifer. It is necessary to continue taking measures for the preservation of water; similarly, it is suggested to continue monitoring the aquifer piezometric levels to update the model based on data availability.

**Keywords:** aquifer modeling; water management; modflow; groundwater; flow simulation





## 1. Introduction

Mathematical modeling applied to groundwater is a fundamental tool to understand the hydrological functioning of the system when it is subjected to different conditions, both natural and anthropogenic [1]. These conditions can be droughts, floods, pumping, etc. [2]. In addition, the main contribution of aquifer modeling is the capability to perform simulations and make predictions. These are carried out with the aim of improving the comprehensive management of groundwater resources. However, it should always be kept in mind that numerical models are tools and that the quality of the simulation results depends on the development of the conceptual model, knowledge of the geological situation, hydrogeological and hydrogeochemical parameters and the data, particularly the initial and boundary values [3,4].

A mathematical model consists of a set of differential equations that are known to govern the flow of groundwater [5]. The approximation of a groundwater model with the real water system determines the reliability of the obtained results; however, to build a model, it is essential to make simplifications because the simulation of the real system is complex, and a simulation result may never be obtained otherwise [6]. Rather, what we obtain is an approximate simulation [7]. Even when a model only approximates the real system, this tool is globally accepted because the cost is considerably lower compared to any other technique used for the qualitative and quantitative analyses of aquifers [8].

Currently there are various types of software that are widely used for modeling and simulating groundwater dynamics; each software uses a mathematical method to represent groundwater equations [9], assumptions and the range of simulation capabilities. In this work, MODFLOW was utilized because of its open-access nature; it was developed by the

USGS [10]. In addition, as a graphic display, Model Muse was used, also developed by the USGS, having free access with high performance capabilities [11].

The problem of water availability for human consumption and agricultural activities is of common knowledge; therefore, the capability of governments and the society, in its entire dimensions, to monitor, manage and make decisions is of primordial importance. In this work, we will focus on the construction of an accurate simulation model of the Celaya Valley aquifer in the Guanajuato State in Mexico, in order to simulate its behavior under certain conditions. This is done since the actual monitoring of the state of the aquifer is expensive and performed (when done) through field studies around every ten years in Mexico, as in other developing countries. Once we obtain a reasonably functioning model, after calibration by comparison with the accessible real-field data, we perform transient state simulations in order to make predictions and contribute to government decisions.

This paper is organized as follows: in Section 2.1, we give a brief description of the study area, from its location to the regional geology that characterizes the Celaya Valley aquifer. In Section 2.2, we present the methodology used for the modeling of the aquifer; we include the hydrogeological and mathematical models and the piezometric analysis in addition to the calculation of the components of the hydrogeological balance. In Section 2.3, we present the steady-state and transient calibration of the model for the dry and rainy seasons, as well as the validation of the calibration results and the sensitivity analysis. In Section 3, we present the results obtained through three simulation scenarios: trend, increase and decrease in pumping. Finally, in Section 4, we give the comments and final conclusions of the results obtained.

## 2. Materials and Methods

### 2.1. Study Area and Regional Geology

The Celaya Valley aquifer (CV) is located at the eastern end of the state of Guanajuato, Mexico (Figure 1), between the parallels 20°20′ and 20°53′ of north latitude and the meridians 100°28′ and 101°06′ of west longitude [12]. According to the Guanajuato State Water Commission (CEAG), it has an approximate area of 2817.19 km$^2$.

The aquifer CV covers the municipalities of Comonfort, Apaseo El Grande, Celaya, Villagrán, Juventino Rosas, Cortázar, Apaseo El Alto and Jaral del Progreso; furthermore, the "Laja river" feeds the aquifer. In Figure 1, you can see the studied area; from the Thiessen polygons, the precipitation and average evaporation in the aquifer were estimated, giving values of 623.38 mm/year and 1963 mm/year, respectively, and an ambient temperature of 18.6 °C. In the CV aquifer, several climates are identified, ranging from semi-warm-subhumid to semi-arid-semi-warm, the most prevalent in the study area being simical-subhumid with rains in summer according to the Köppen classification.

Within the area under study there are two physiographic provinces—Neovolcanic Transverse Axis (ENT) and Mesa Central—according to the National Institute of Statistics and Geography (INEGI, its Spanish acronym). The first is characterized by variations in relief and rock types; on the other hand, the Mesa Central has extensive plains that are interrupted by mountain ranges [13].

The Celaya Valley aquifer is located in the "Lerma-Santiago" RH No. 12 Hydrological Region, "La Laja" Hydrological Subregion, in the Lerma-Salamanca River basin, Pericos sub-basin; Irrigation District No. 085, "La Begoña" is located there, which takes advantage of the runoff from the Laja river, controlled by the Ignacio Allende dam, which is located north of the aquifer [12,13]. The main surface currents that drain the studied area come from the Laja river, and also receive as tributaries the Querétaro River and the Nautla Stream. The Laja river empties into the Lerma river, which circulates at the southern end of the CV aquifer.

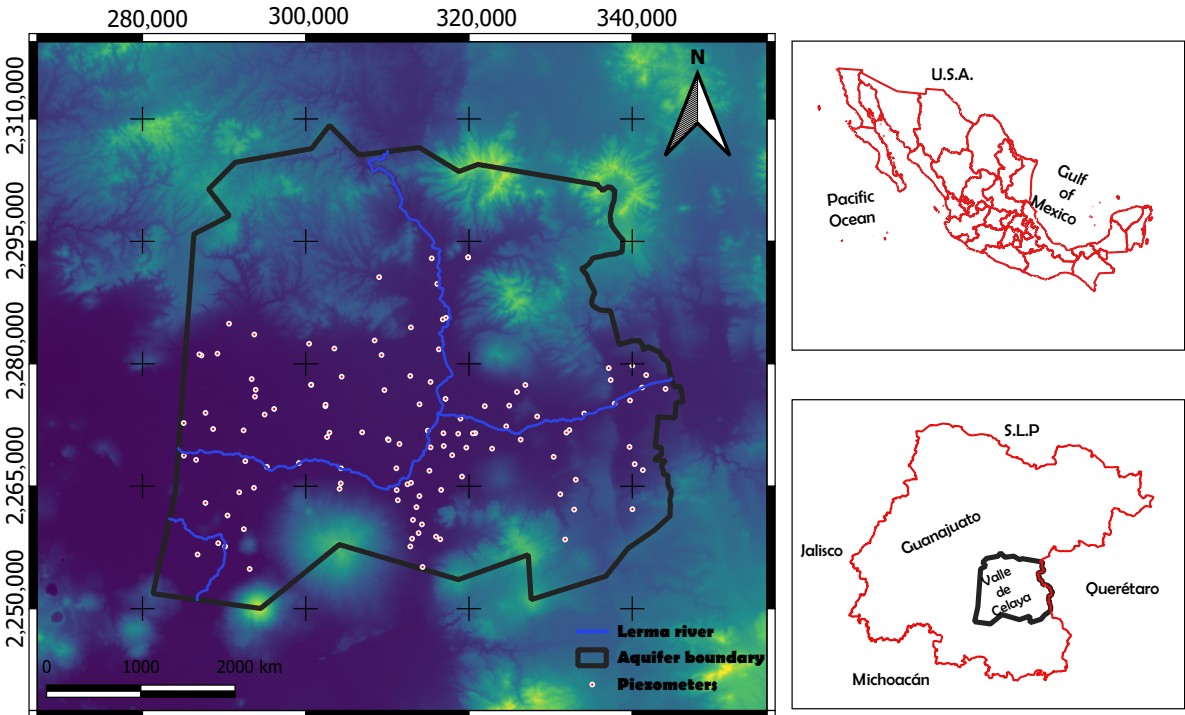

**Figure 1.** Location of the Celaya Valley aquifer, Guanajuato, Mexico. Topographic map of the study area with the location of the piezometric wells. Coordinate system: WGS 84/UTM Zone 14 N.

The various geological maps that frame the study area expose the surface geology of the Celaya Valley aquifer, which is located largely on Neogene–Quaternary sandstone–polygenic conglomerate; the rest corresponds to andesite–basalt from the Neogene, Quaternary and Tertiary. In addition, there is rhyolite-rhyolitic tuff from the Paleogene and Tertiary. In smaller proportions, there is the presence of limestone–siltstone with Cretaceous volcanic–sedimentary rocks, as well as Tertiary andesite [14]. In summary, in the region where the aquifer in question is located, mainly volcanic and sedimentary rocks outcrop, whose stratigraphic records suggest the ages of these rocks as being between the Cretaceous and Quaternary (Figure 2).

*2.2. Methodology*

2.2.1. Hydrogeological Model

The geological and geophysical studies carried out in the CV aquifer indicate that this geological formation, due to its hydrodynamic behavior, is a free-semi-confined aquifer, of a granular and fractured type, heterogeneous and anisotropic. The upper part of the aquifer is constituted by alluvial sediments, sandstones and conglomerates. Its granulometry varies from clay to gravel, whose thickness usually reaches hundreds of meters. On the other hand, the lower portion of the aquifer is constituted by fractured volcanic rocks of basaltic and rhyolitic composition.

The layered model of this aquifer is constituted by four layers (Figure 3): the first corresponds to the storage unit formed by sandstones; the second layer is defined from andesitic and basaltic rocks, in which the permeability is high due to their great fracturing; the third layer is formed by a unit of rhyolitic rocks; and the last layer belongs to the conglomerates, which is considered the boundary condition because of its low hydraulic conductivity.

# Geology

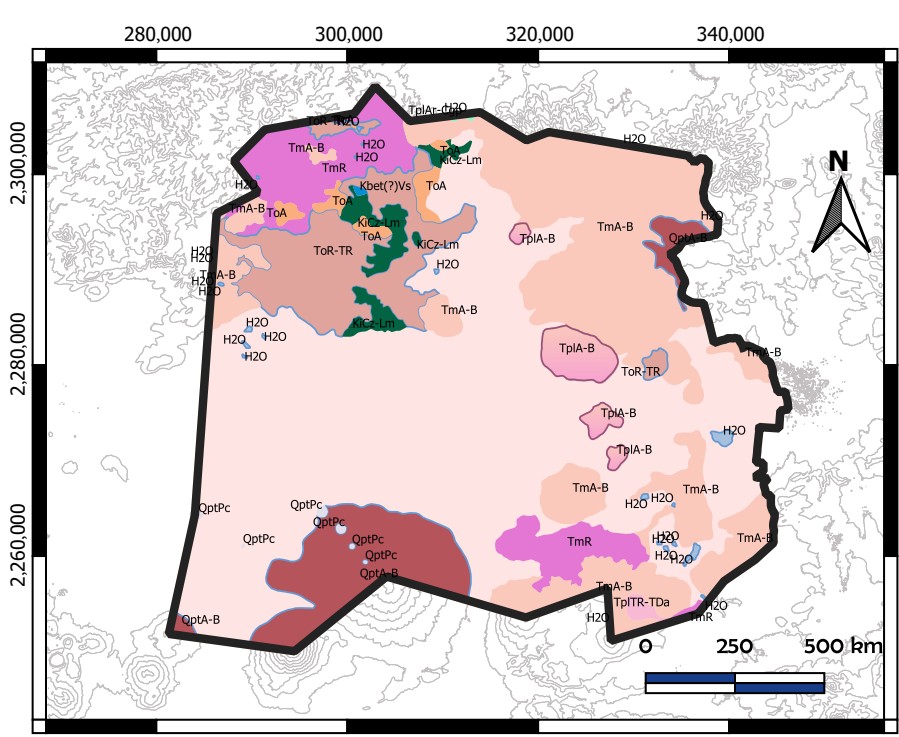

## SYMBOLOGY

**CENOZOIC**

**Quaternary**

- Pyroclastic (QptPc)
- Andesite - Basalt (QptA-B)

**Neogene - Quaternary**

- Sandstone - Polygenic conglomerate

**Neogene**

- Andesite - Basalt (TplA-B)
- Rhyolitic tuff - Dacitic tuff (TplTR-TDa)

- Andesite - Basalt (TmA-B)

**Tertiary**

- Andesite (ToA)
- Rhyolite - Rhyolitic tuff (ToR-TR)
- Miocene rhyolite (TmR)

**MESOZOIC**

**Cretaceous**

- Volcanic sedimentary (Kbet(?)Vs)
- Limestone - Limonite (KiCz-Lm)

**Figure 2.** Geology map of the CV aquifer. Coordinate system: WGS 84/UTM Zone 14 N.

## 2.2.2. Static Level

The static level was measured in two seasons: dry and rainy. The first covers the months of August to October, and the second seasons covers the months of March to May. As is known, the elevation of the static level is obtained by measuring the depth of the static level, the distance from the surface to the water level. This depth is obtained using a probe that is inserted into each well; the measured distance is subtracted from the height of the point above the mean sea level, and the result is the elevation of the static level. It should always be considered that the topography of an area is not uniform; each well has a different height and to obtain this height, the curb leveling is carried out by placing a GPS receiver in each well, obtaining the height to which the height of the curb is added,

then the depth of the static level is subtracted. In this way, the elevation of the static level is obtained [15].

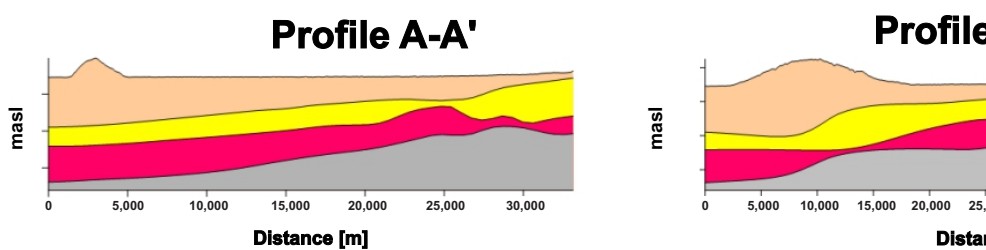

**Figure 3.** Hydrogeological model of the CV aquifer, Profile A-A' corresponds to a geological section from South to North and Profile B-B' corresponds to a geological section from West to East.

To perform the analysis of the behavior of the static level, the data for 2010 and 2015 were consulted for the rainy season, while for the dry season, those for the years 2015 and 2019. In addition, with these data, the depth and elevation maps of the static level were constructed, which we present below.

Dry Season

Figure 4a shows the static level depth map for the year 2015; in this year, it can be seen that to the east of Apaseo el Alto, there are increasing depths with an east–west direction ranging from 60 m to 120 m. Once they cross the entire municipality of Comonfort, the depths begin to decrease, since a small cone can be seen to the south of Celaya, which reaches a depth of 90 m, in addition to another cone formed in the center of the city of Celaya, with a depth of 120 m. A more open cone to the west of Celaya with depths of 60 m is also highly visible. Between Celaya, Cortázar, Villagrán and Juventino Rosas, in the same way, a marked cone is visualized that reaches depths of 150 m; finally, to the northeast of Jaral del Progreso, the depths improve from 30 m to 60 m.

On the other hand, the static level elevation map of the year 2015 (Figure 4b) allows to view three possible water inlets to the aquifer; the first is located from Apaseo el Grande to Celaya; however, it feeds a cone formed between Celaya, Juventino Rosas, Villagrán and Cortázar, being defined by an equipotential line of 1650 m above sea level (masl). A second input is seen to the east of Jaral del Progreso, where a small cone is also formed to the southwest of Villagrán and Cortázar. The last input is marked from Apaseo el Alto to the municipality of Comonfort, and in the same way, it feeds an extensive cone to the east of Celaya that reaches an equipotential line of 1600 masl.

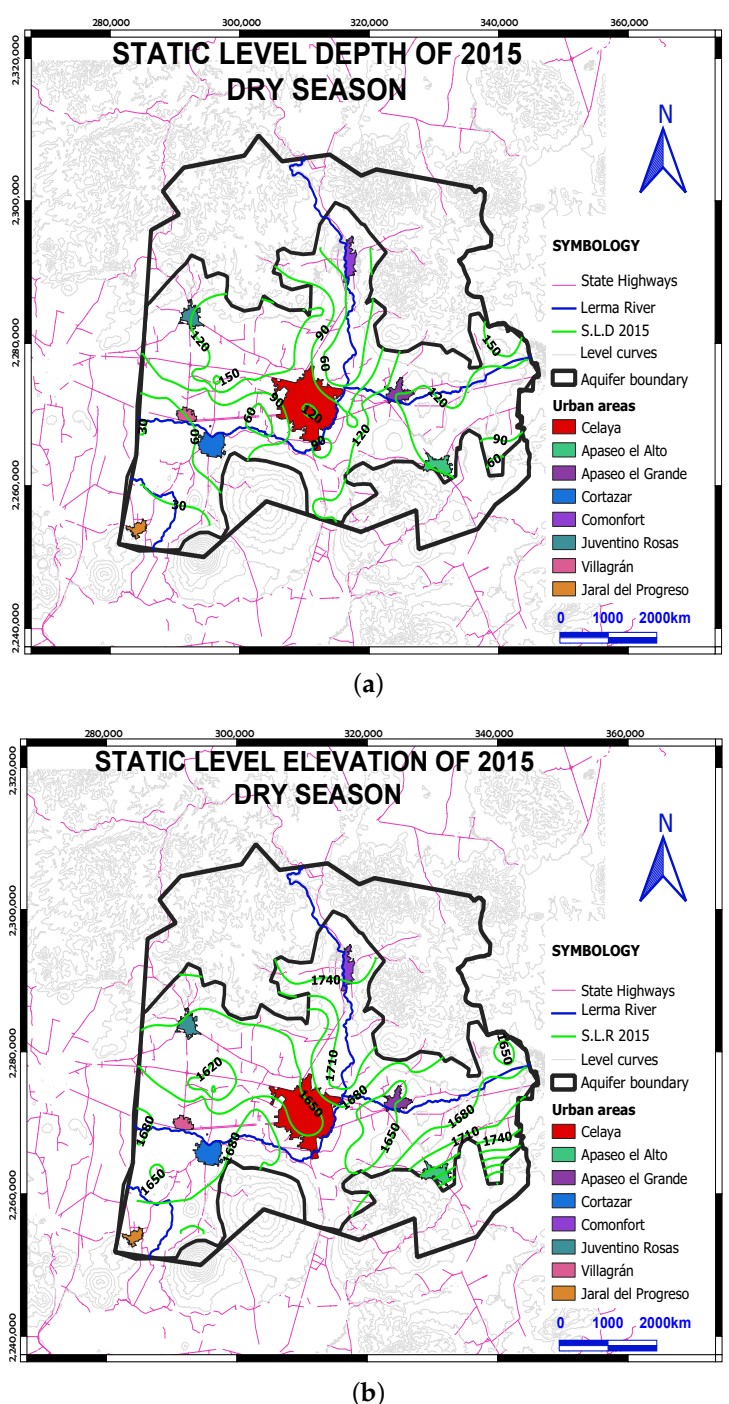

**Figure 4.** Dry season of 2015. (**a**) Distribution of static level depth lines. (**b**) Distribution of static level elevation lines.

In relation to the depth of the static level in 2019, several abatement cones are observed, mostly on the west side of Celaya. From the municipality of Jaral del Progreso, the depths begin to increase toward the municipality of Cortázar from 30 m to 90 m. Between the municipalities of Celaya, Juventino Rosas, Villagrán and Cortázar, there is a series of very closed cones that reach depths of up to 120 m. Following that, toward Apaseo el Grande, Comonfort and Celaya, a large cone is formed that extends with depths of 90 m in the center to 120 m in these municipalities; see Figure 5a.

The elevation of the static level in 2019 allows us to observe the three inputs that are visualized in the year 2015, one of them located in Apaseo el Alto, where the flow continues

its course toward the municipalities of Celaya and Apaseo el Grande until meeting the cone generated in Apaseo el Grande, which covers the entire municipality and is defined by the equipotential line of 1650 m above sea level. It is also possible to observe in the center of the municipality of Celaya the formation of a cone that covers the entire municipality and is defined by the equipotential line of 1650 m above sea level. A second input is observed to the east of Jaral del Progreso; the flow is directed northeast and continues its course until the cone located in Celaya. Another input is located in Apaseo el Grande. The flow goes south until it reaches the Celaya and Comonfort cones; see Figure 5b.

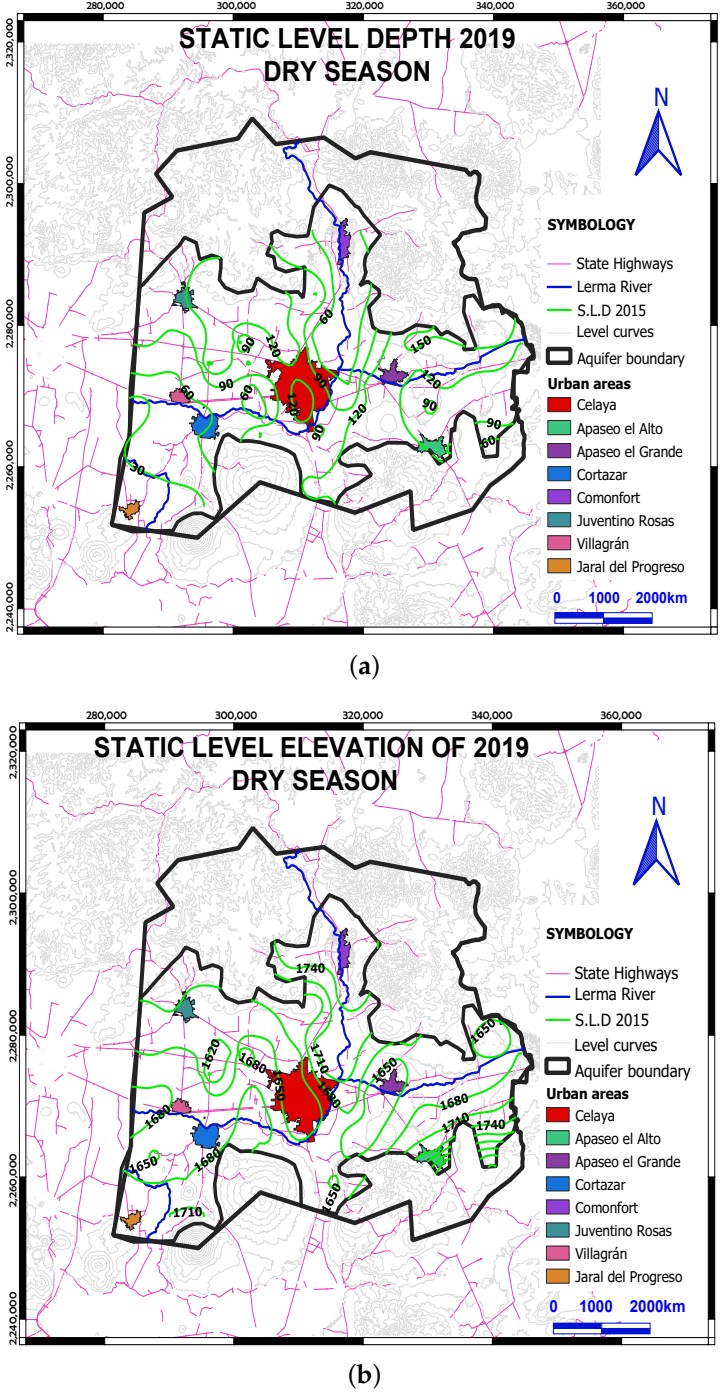

**Figure 5.** Dry season of 2019. (**a**) Distribution of static level depth lines. (**b**) Distribution of static level elevation lines.

Rainy season

Figure 6a shows the static level depth map for the year 2010. There are four abatement cones: a first cone located southeast of Juventino Rosas that extends up to 90 m; a second cone formed between the municipalities of Apaseo el Grande and Celaya that reaches a depth of 30 m; and the other two cones are located east and west of Celaya with depths of 60 m and 90 m, respectively. In the northeast of Jaral del Progreso, the depth begins to increase from 30 m to 90 m, which is part of the feeding of the small cone located northwest of Celaya.

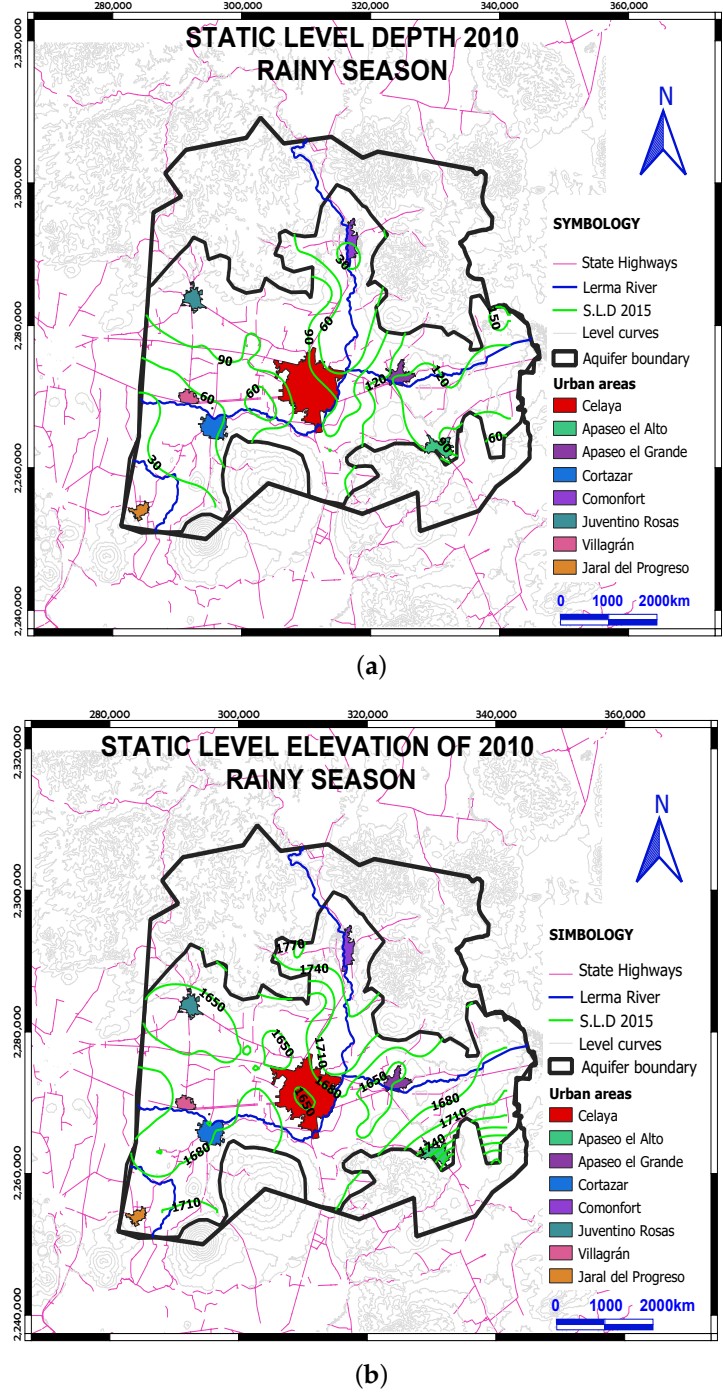

**Figure 6.** Rainy season of 2010. (**a**) Distribution of static level depth lines. (**b**) Distribution of static level elevation lines.

In year 2010, the static level elevation configuration shows that the three possible water inlets to the aquifer mentioned in the dry season continue to be present, the first coming from the north of the aquifer, near Apaseo el Grande, where the flow continues until it reaches the municipality of Celaya (Figure 6b). The other input comes from the southeast of the aquifer. The equipotential lines go from 1700 m above sea level to 1680 m above sea level; however, near the municipality of Comonfort and Celaya, they feed an extensive cone of 1650 m above sea level. Another of the possible inputs continues to be from the east of the municipality of Jaral del Progreso, where the flow direction is toward the north, which also feeds a cone. Located west of Cortázar, there are also small cones in Celaya, one in the center, and another two to the northwest and southwest.

In relation to the depth of the static level in 2015, various cones are observed to be distributed throughout the aquifer, the largest being located between the municipalities of Celaya, Apaseo el Grande and Comonfort whose depth reaches 40 m; in turn, this cone joins a second cone located southeast of the aquifer with a depth of 60 m. To the west of Villagrán, there are relatively shallow depths; to the southwest of Juventino Rosas, the formation of a cone with a depth of 80 m is visualized. The municipality of Apaseo el Alto has the greatest depths reaching 160 m; see Figure 7a.

Figure 7b shows the static level elevation configuration of the year 2015, where three recharge zones that appear in the dry season are also present: One of them comes from the southeast border of the aquifer, where the flow moves toward the municipalities of Celaya and Comonfort; there is a large cone defined with the equipotential line of 1650 masl and presents a soft elongation towards the northeast of Celaya. The second input comes from the east of Jaral del Progreso, which follows a northeast direction passing through Cortázar and Celaya, reaching the cone that is generated between Celaya, Juventino Rosas and Cortázar and is defined by the equipotential line of 1620 masl. Finally, another observed input comes from the Apaseo el Grande municipality, where the flow is directed to feed the Celaya cone.

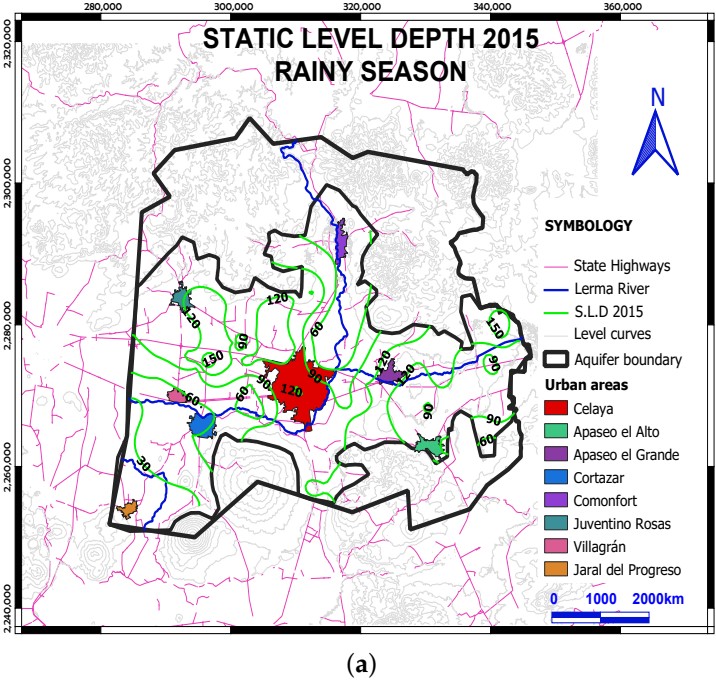

(**a**)

**Figure 7.** *Cont.*

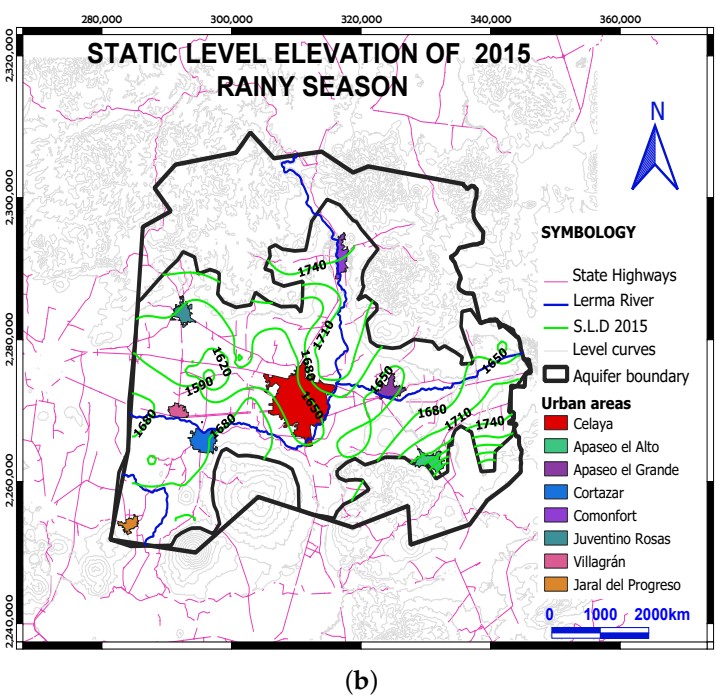

**(b)**

**Figure 7.** Rainy season of 2015. (**a**) Distribution of static level depth lines. (**b**) Distribution of static level elevation lines.

### 2.2.3. Static Level Evolution

In order to carry out a history of the piezometric behavior in the Celaya Valley aquifer, the evolution of the static level was estimated for both the dry and rainy seasons from 2015 to 2019, while for the other, it was from 2010 to 2015. The procedure for calculating the evolution of the static level consists of the subtraction between the elevation of the static level of the piezometric data of the oldest date, minus the elevation of the static level of the data with more recent date, and whose result between these two periods shows the change in the levels. If the result is negative, it means that the aquifer receives recharge, while if the change is positive, this indicates that the aquifer is dejected [16,17].

For the configuration of the evolution of the static level of the dry season, 78 wells were used. In general terms, there are significant recoveries. In some parts of the Valley, there is neither a loss nor gain, but to the north of the municipality of Celaya, it is possible to see a great loss of the static level of at least 10 m; see Figure 8a.

In the rainy season, 73 wells were used to know the evolution of the static level. There are important losses to the west of the municipality of Celaya, which correspond to almost half the Valley, that reach up to 20 m. On the other hand, to the east of Celaya, there are recharges of 5 m; these values are not high, but they guarantee that in this time interval, there was the recovery of some areas of the aquifer. It is also possible to appreciate that the aquifer is fed by the Laja river since, although there are not so many gains, nor are there losses in its journey; see Figure 8b.

### 2.2.4. Groundwater Balance

According to CEAG for the year 2000, originally, the underground flow in the aquifer had a direction from east to west [18], that is, from Celaya to the municipalities of Villagrán and Salamanca; however, at present, the large extractions of groundwater in the aquifer have caused a deformation of a piezometric cone southeast of Juventino Rosas and northeast of Villagrán, the site toward which the flows are directed.

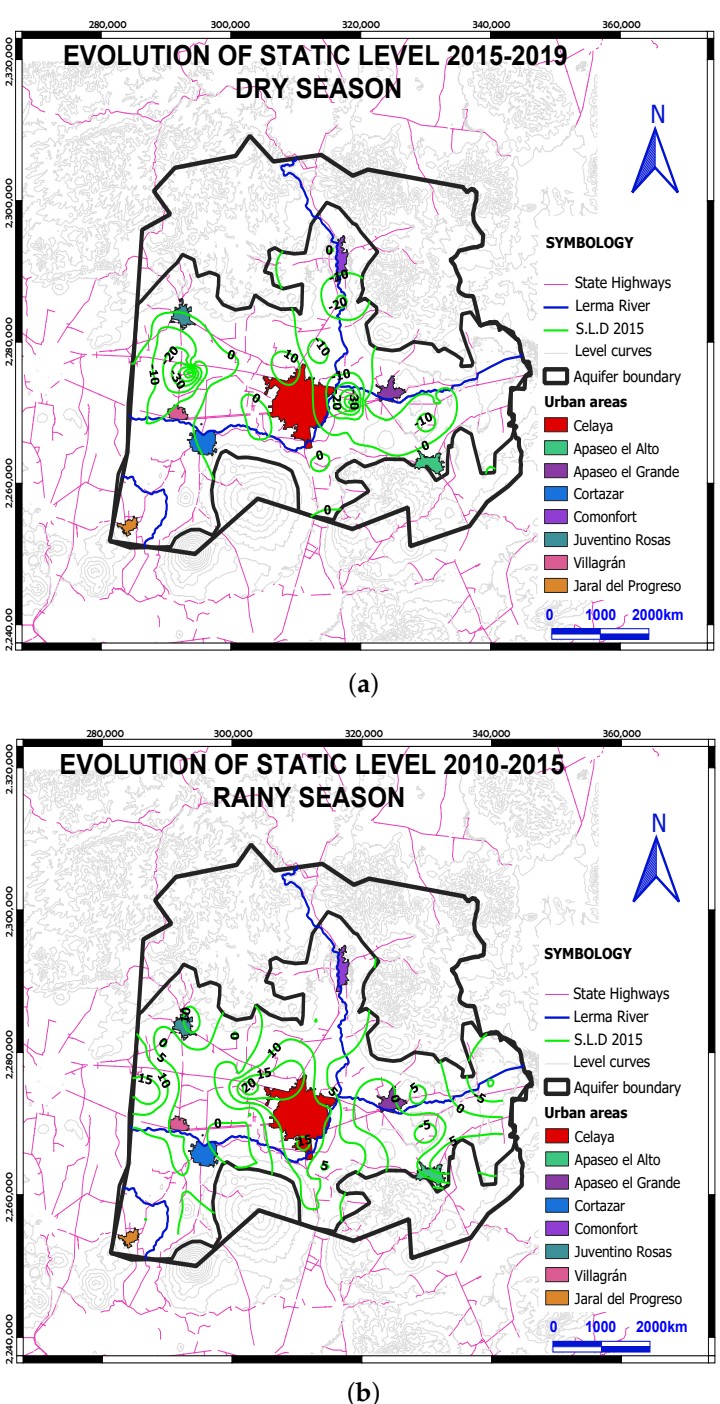

**Figure 8.** Static level evolution. (**a**) Analysis period of 2015 to 2019 in dry season. (**b**) Analysis period of 2010 to 2015 in rainy season.

The aquifer receives natural recharge from the topographically high areas, such as the elevations of the "La Gavia" volcano, "Culiacán" and the "Sierra Codornices", in the same way, from the "Laja" river and by the infiltration of surface waters as part of horizontal inlets. In addition, a part of the volume of water that enters the aquifer is due to return through irrigation and urban public use [12]. The greatest amount of underground flow is available from the North Sierra to the center of the Valley and from the South Sierra also toward the Valley. The extractions of water in the aquifer are carried out mainly by pumping from a large number of wells that extend throughout the Valley. Most of these wells are for agricultural, urban and industrial use; evapotranspiration is another outlet

source. Table 1 summarizes the water inlets and outlets of this aquifer system for the dry season and in Table 2 for the rainy season.

For the calculation of the horizontal inputs, the static level elevation configuration for the dry season of 2019 was considered because this year, it presented a greater amount of data to completely cover the studied area. From Darcy's law [19], the flow $Q$ was calculated and for this purpose, 16 wells were used. The sums of these flows allowed a total volume of underground flow to be counted that amounts to 175.3 mm$^3$/year. The transmissivity values used for this calculation were obtained from the average of the pumping tests carried out as part of a CONAGUA study in 2016 and other previous studies, adapted to the saturated thickness of each zone.

In the case of natural recharge, the CLICLOM database of 22 climatological stations with influence on the aquifer was used [20,21]; these were Ameche, Apaseo El Alto, Apaseo El Grande, Celaya (SMN), Celaya (DGE), Cortázar, Two Streams, El Obraje, El Sabino, La Begoña, El Terrero, Neutla, Pericos, Jalpa Dam, Roque, Santa Rita, Valle de Santiago, Tres Guerras, Comonfort, La Joyita, El Pueblito and Juriquilla. These stations work independently and have acquired data during different time periods, due to the moment each station began to work, the broadest being 1922–2016 and the shortest 1980–2016, from which we take the 1980–2016 period since it provides data for all 22 stations. Working out the calculations, a precipitation volume of 7284 mm$^3$ was obtained, for which the average precipitation sheet was 623.38 [mm].

To calculate the volume of runoff in the aquifer, the hydrometric stations surrounding the study area were located from the database of the National Bank of Surface Waters BANDAS (IMTA) [22], giving a total of four hydrometric stations within the limit of the aquifer; the runoff value was 90.18 mm$^3$.

**Table 1.** Variables for calculating the groundwater balance in the Valle de Celaya aquifer in the dry season.

| | | | |
|---|---|---|---|
| Natural recharge by rain | $Rv$ | mm$^3$/year | 161.0 |
| Horizontal inputs | $Eh$ | mm$^3$/year | 175.3 |
| TOTAL NATURAL RECHARGE | | mm$^3$/year | 336.3 |
| Return for public-urban use | $Rv$ | mm$^3$/year | 5.7 |
| Return by irrigation (groundwater) | $Ev$ | mm$^3$/year | 82.1 |
| Return by irrigation (surface water and waste) | $Rv$ | mm$^3$/year | 20.2 |
| TOTAL RETURN | | mm$^3$/year | 108 |
| Recharge by river (conduction losses) | | mm$^3$/year | 20.7 |
| Recharge by waterway (conduction losses) | | mm$^3$/year | 3.3 |
| **TOTAL RECHARGE** | **Inputs** | **mm$^3$/year** | **468.3** |
| Agricultural | | mm$^3$/year | 462.5 |
| Public-urban | | mm$^3$/year | 71.1 |
| Industrial | | mm$^3$/year | 26.7 |
| Others | | mm$^3$/year | 20.0 |
| TOTAL GROSS EXTRACTION | | mm$^3$/year | 580.3 |
| Evapotranspiration | | mm$^3$/year | 58.6 |
| **TOTAL DISCHARGE** | **Outputs** | **mm$^3$/year** | **638.9** |
| **INPUTS—OUTPUTS** | | **mm$^3$/year** | **−170.6** |

In order to calculate the infiltration volume, the triangular unit hydrogram (HUT) methodology was applied to obtain the infiltration sheet [23]. The total runoff volume gave a value of 93.53 mm$^3$/year; therefore, the infiltration sheet obtained by this method is 0.8578 m. To obtain this value, the difference between the precipitation volume and the runoff volume was calculated, and later it was divided by the study area.

In the case of return by irrigation and urban public use, the Irrigation District No. 85 "La Begoña" was used [24], which covers an area of 117.0222 km$^2$, and according to the infiltration of this aquifer, a vertical recharge was obtained by return from surface water irrigation of 20.2 mm$^3$/year and a horizontal return from groundwater irrigation of 82.1 mm$^3$/year. In addition, the volume of effective recharge in the aquifer through the return by drinking water distribution networks due to leaks was 5.7 mm$^3$/year, giving a total value for returns of 108.0 mm$^3$/year.

**Table 2.** Variables for calculating the groundwater balance in the Valle de Celaya aquifer in the rainy season.

| | | | |
|---|---|---|---|
| Natural recharge by rain | $Rv$ | mm$^3$/year | 178.3 |
| Horizontal inputs | $Eh$ | mm$^3$/year | 201.5 |
| TOTAL NATURAL RECHARGE | | mm$^3$/year | 379.8 |
| Return for public–urban use | $Rv$ | mm$^3$/year | 5.7 |
| Return by irrigation (groundwater) | $Ev$ | mm$^3$/year | 82.1 |
| Return by irrigation (surface water and waste) | $Rv$ | mm$^3$/year | 20.2 |
| TOTAL RETURN | | mm$^3$/year | 108 |
| Recharge by river (conduction losses) | | mm$^3$/year | 25.1 |
| Recharge by waterway (conduction losses) | | mm$^3$/year | 4.2 |
| **TOTAL RECHARGE** | **Inputs** | **mm$^3$/year** | **517.1** |
| Agricultural | | mm$^3$/year | 462.5 |
| Public–urban | | mm$^3$/year | 71.1 |
| Industrial | | mm$^3$/year | 26.7 |
| Others | | mm$^3$/year | 20.0 |
| TOTAL GROSS EXTRACTION | | mm$^3$/year | 580.3 |
| Evapotranspiration | | mm$^3$/year | 55.1 |
| **TOTAL DISCHARGE** | **Outputs** | **mm$^3$/year** | **635.4** |
| **INPUTS–OUTPUTS** | | **mm$^3$/year** | **−118.3** |

### 2.2.5. Mathematical Model

Currently, the most common and used methods for modeling dynamics in aquifers are the finite difference method (FDM) [25] and the finite element method (FEM) [5]. To model the Celaya Valley aquifer, MODFLOW-2005 was used. This software solves the groundwater flow equation in a three-dimensional way by the finite difference method, which uses a rectangular mesh where an approximation of the first derivatives of the partial derivative equations is applied as a quotient of differentials [26].

The groundwater equation of motion, derived experimentally in the form of Darcy's law [27,28], is flow-limited in one dimension; therefore, we consider the analysis of the dynamics of the underground flow through a three-dimensional system with coordinates $(x, y, z)$, then the generalization of Darcy's equation [29] in 3D is

$$\vec{q} = -\sigma \cdot \nabla h(x, y, z), \tag{1}$$

where $\vec{q}$ is the specific discharge vector and $\nabla h$ is the hydraulic gradient with components $\frac{\partial h}{\partial x}$, $\frac{\partial h}{\partial y}$, $\frac{\partial h}{\partial z}$, in the $x$, $y$ and $z$ directions respectively, and the tensor $\sigma$ is considered as diagonalizable to three principal directions, that is, it has three components on the axes of the Cartesian coordinates. This symmetric tensor is called the hydraulic conductivity tensor, and in terms of its components, it can be written as

$$\sigma = \begin{pmatrix} \sigma_{xx} & \sigma_{xy} & \sigma_{xz} \\ \sigma_{yx} & \sigma_{yy} & \sigma_{yz} \\ \sigma_{zx} & \sigma_{zy} & \sigma_{zz} \end{pmatrix},$$

then Equation (1) can be written in terms of three equations, one term for each direction

$$
\begin{aligned}
q_x &= -\sigma_{xx}\frac{\partial h}{\partial x} - \sigma_{xy}\frac{\partial h}{\partial y} - \sigma_{xz}\frac{\partial h}{\partial z} \\
q_y &= -\sigma_{yx}\frac{\partial h}{\partial y} - \sigma_{yy}\frac{\partial h}{\partial y} - \sigma_{yz}\frac{\partial h}{\partial y} \\
q_z &= -\sigma_{zx}\frac{\partial h}{\partial z} - \sigma_{zy}\frac{\partial h}{\partial y} - \sigma_{zz}\frac{\partial h}{\partial z}
\end{aligned}
\tag{2}
$$

where $\sigma_{i,j}$ correspond to the components of the hydraulic conductivity tensor.

In order to obtain the scalar field $h$ from Equation (1), the divergence of this equation is taken so that it reflects the dynamic behavior of water, obtaining

$$\nabla \cdot (\sigma \cdot \nabla h) = 0, \tag{3}$$

and therefore, the expansion into components of Equation (3) is

$$\frac{\partial}{\partial x}\left(\sigma_{xx}\frac{\partial h}{\partial x}\right) + \frac{\partial}{\partial y}\left(\sigma_{yy}\frac{\partial h}{\partial y}\right) + \frac{\partial}{\partial z}\left(\sigma_{zz}\frac{\partial h}{\partial z}\right) = 0. \tag{4}$$

When Equation (4) is combined with the water balance equation in a small control volume [30], for the case when sources or wells are present, Darcy's Law leads to a partial differential equation that describes the hydraulic head distribution:

$$\frac{\partial}{\partial x}\left(\sigma_{xx}\frac{\partial h}{\partial x}\right) + \frac{\partial}{\partial y}\left(\sigma_{yy}\frac{\partial h}{\partial y}\right) + \frac{\partial}{\partial z}\left(\sigma_{zz}\frac{\partial h}{\partial z}\right) + Q'_s = SS\frac{\partial h}{\partial t}, \tag{5}$$

where $Q'_s$ $[T^{-1}]$ is the volumetric flow rate per unit volume representing sources and sinks of water, being negative for the flow out of the groundwater system and $Q'_s$ positive for the incoming flow ; $SS$ corresponds to the specific storage of the porous material $[L^{-1}]$ and t is the time $[T]$. In general, $SS$, $\sigma_{xx}$, $\sigma_{yy}$, and $\sigma_{zz}$ tend to be functions of space, while $Q'_s$ can be a function of both space and time [31].

In general, Modflow 2005 has the ability to model both steady-state and transient systems [32]. Hydrogeological layers can also be defined as free, confined and semi-confined, in addition to simulating stresses in the system by adding wells, drains, evapotranspiration, recharge areas, rivers, among others. Additionally, values of hydraulic conductivity and storage coefficient are declared, in each one of the directions of space, where the values can be different for each layer.

Spatial and Temporal Discretization

The Celaya Valley aquifer is defined between a series of mountain ranges and dome-shaped mountains [33], it is a plain of lacustrine and alluvial sediments that is found with an elevation below 1850 meters above sea level. In this way, it was defined that the area to be modeled has an approximate surface of 1256 km$^2$. This is distributed in a mesh of 65 cells in the $X$ direction and 59 cells in the $Y$ direction, being a total of 3835 square cells

with dimensions of 1 km per side; of this total, only 1624 cells are active in the simulations process. The rest of the cells are inactive because they correspond to the main topographic elevations; see Figure 9.

On the other hand, in the vertical (*z*) direction, we worked with four layers, which are specified in Section 2.2.1, the hydrogeological model. Of the four layers in the model, the first was treated as convertible [34] due to the semi-confined character of the aquifer and the remaining three layers as confined. Regarding the modeling time, a stress period with a duration of five years was established for the dry and rainy season.

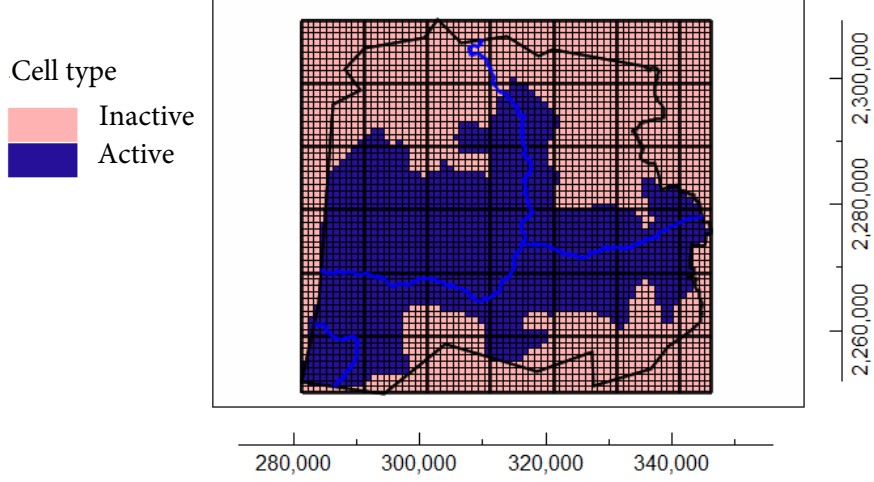

**Figure 9.** Spatial discretization, plan view of the model.

According to the distribution and shape of the graben present in the aquifer, the granulometry of the alluvial materials varies from clay to gravel, and its thickness reaches several hundred meters; on the other hand, the fractured volcanic medium is very thick, with heterogeneous–anisotropic properties due to the fracturing pattern that determines the occurrence and distribution of groundwater.

For the modeling, the distribution of active cells from the first layer to the last is preserved; however, the communication that exists in each of the aquifer layers is respected, making a distinction between these layers only by the variations of the hydrogeological parameters.

Initial and Boundary Conditions

The modeled area of the aquifer was limited by lake and alluvial sediments, by following the 1850 masl level curve, the limit of the aquifer was restricted to a valley. Likewise, rivers were considered as internal borders or flow. Due to the fact that there is a plain surrounded by mountains and hills, the flow direction follows its natural tendency from higher to lower topography [35]; the contributions from the highest elevations discharge toward the Celaya Valley.

To represent the natural hydrological aspects of the aquifer, MODFLOW uses different packages [8]. The ones used in this model were RIV to represent each section of the Laja River and Lerma, RCH to enter the value of the recharge sheet, GHB to designate the horizontal inputs to the system, EVT to express the outputs by evapotranspiration, and DRN to represent the value of the irrigation channel contribution.

Hydrogeological Parameters

The hydrogeological aspects that are part of the modeled system are the hydraulic conductivities, both horizontal and vertical [meters/day], and the specific coefficient [dimensionless]. The lateral borders were considered as wells that simulate the recharge by underground flow deduced from the cells, as well as extraction wells that correspond to the outlet by underground flow in the Valley.

With respect to the horizontal conductivities, these were assigned according to the values thrown by the pumping tests and by the type of existing materials; reported by Lesser [18], this parameter varied from 2.5 to 5.0 m/day. In the case of vertical conductivity, the relationship equivalent to 10% of horizontal conductivity was preserved, as reported by Freeze [29].

*2.3. Calibrations*

2.3.1. Stationary State Calibration

For the simulation in steady state, it is established that the system to be modeled remains in equilibrium since the time variable is not involved [36,37], that is, the hydraulic heads remain unchanged because there are no external factors that intervene in the process. In this case, the elevation of the static level of the year 2015 was considered the initial head, and the established time was one day; in addition, it was sought to reproduce the values of the hydraulic heads measured in the field. The calibrated parameters were groundwater inflows (GHB), river inflows (RIV), recharge (RCH), channels (DRN), and evapotranspiration (ET).

The calibration was carried out using the trial-and-error method; mainly the conductance in the river was varied since it was one of the parameters with the greatest uncertainty, and the pumping effect was not considered [38].

Dry Season

The calibration considers the mass balance provided in the results of the model run as percent discrepancy, in this case, zero. In addition, Table 3 shows the value of the calculated and simulated parameters, as well as the calculation of the percentage absolute error for the simulated values. Finally, in Figure 10, the result of the hydraulic head is presented, where the direction of flow toward the Valley is displayed; this is associated with the fact that it is the area with the lowest topographic elevation.

**Table 3.** Percentage of error for the calibration in steady state of the dry season.

| PARAMETER | CALCULATED [$m^3$/Year] | SIMULATED [$m^3$/Year] | ERROR % |
|:---:|:---:|:---:|:---:|
| RECHARGE | 736,986.30 | 732,501.44 | 0.61 |
| RIVER LEAKAGE | 56,712.33 | 56,971.26 | 0.46 |
| GHB | 480,273.97 | 484,238.19 | 0.83 |
| DRAINS | 9041.10 | 9450.52 | 4.53 |
| ET | 160,547.95 | 162,032.87 | 0.92 |

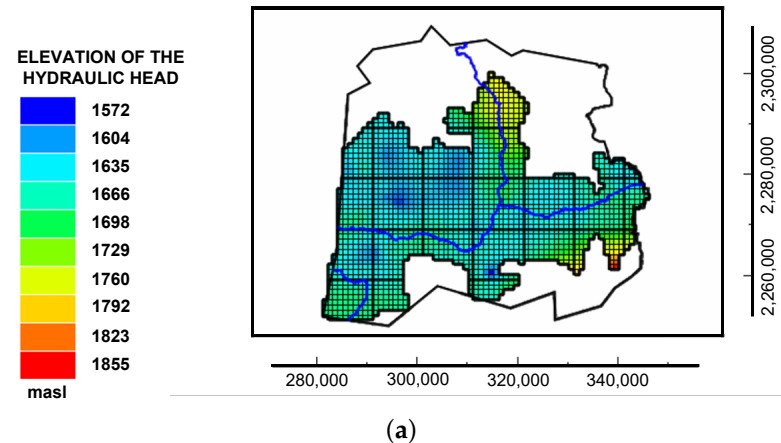

(**a**)

**Figure 10.** *Cont*.

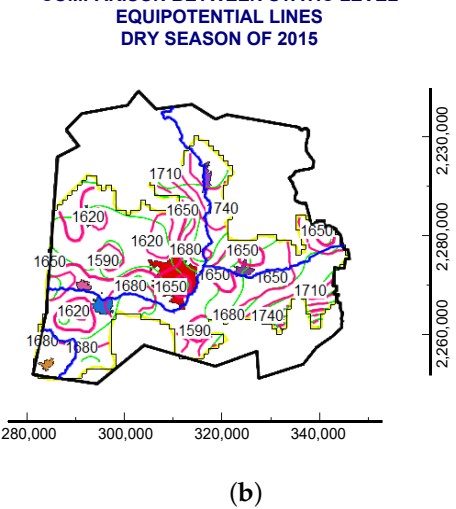

(**b**)

**Figure 10.** Dry season. (**a**) Hydraulic head in steady state, where the highest head is represented by red and the lowest head by dark blue. (**b**) Comparison between the equipotential lines of the piezometric levels calculated by the model (pink color) and those measured in the field (green color).

Rainy Season

For this season, the same water balance was considered as in the dry season. Also in this case, the calibration for this season considers the mass balance with a percent discrepancy of zero. Table 4 shows the value of the calculated and simulated parameters and the absolute error for the values obtained. In addition, in Figure 11, the resulting hydraulic heads is presented, and in the same way as in the dry season, the flow direction is toward the Valley.

**Table 4.** Percentage of error for the calibration in steady state of the rainy season.

| PARAMETER | CALCULATED [$m^3$/Year] | SIMULATED [$m^3$/Year] | ERROR % |
|:---:|:---:|:---:|:---:|
| RECHARGE | 784,383.56 | 776,879.24 | 0.96 |
| RIVER LEAKAGE | 68,767.12 | 69,445.22 | 0.99 |
| GHB | 552,054.79 | 547,311.89 | 0.86 |
| DRAINS | 11,506.85 | 10,779.52 | 6.32 |
| ET | 150,958.90 | 152,132.87 | 0.78 |

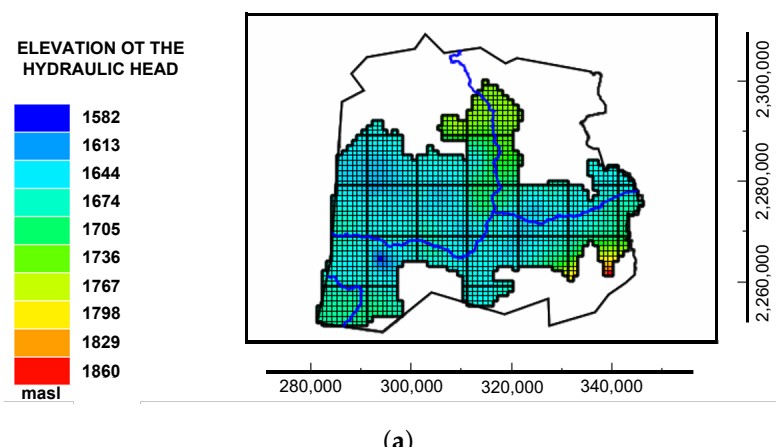

(**a**)

**Figure 11.** *Cont.*

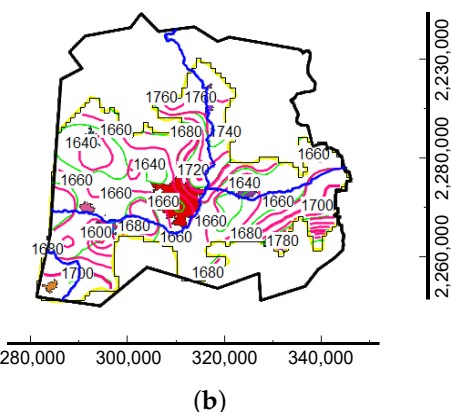

**(b)**

**Figure 11.** Rainy season. (**a**) Hydraulic head in steady state, where the highest head is represented by red and the lowest head by dark blue. (**b**) Comparison between the equipotential lines of the piezometric levels calculated by the model (pink color) and those measured in the field (green color).

### 2.3.2. Calibration in Transient State

The transient state is characterized by the change in hydraulic heads over time, that is, there is a disturbance of the natural functioning of the system. The extraction of water by pumping to supply the agricultural sector is characteristic of this aquifer; however, these massive extractions have caused a great decrease in water storage, subjecting the system to a state of stress. For the transient state simulations, the initial heads were defined, the storage data were fed in, and the extraction wells were activated.

### Dry Season

As initial conditions, the hydraulic heads of the year 2015 were established; therefore, a configuration of the initial head was made with the elevations derived from the measurements in the field, and the configuration that was sought to be reproduced corresponds to the year 2019. The simulation period was 4 years, equivalent to 1461 days.

For the transient state simulation, the model uses a specific storage coefficient *SS* of $8 \times 10^{-6}$ in the first and second layers and $3.2 \times 10^{-5}$ in the rest of the layers; these values totally depend on the lithology. In the case of extraction volumes, a file was created that included the wells that extract water from the aquifer, and whose operating regime is known. The value of the extraction flows was variable, reaching a maximum value of 14,800 m$^3$/day.

To carry out the calibration in a transitory state, it was necessary to adjust the conductance of the Lerma river and the hydraulic conductivity of the model layers, since in this way, it was possible to reproduce the flow directions in the aquifer and with it, the levels of the expected hydraulic heads. It is worth mentioning that this calibration was carried out using the trial-and-error method; Figure 12 shows the hydraulic head obtained by the model for the period 2015–2019.

Considering that discrepancies between observed and calculated responses of a system are the manifestation of errors in the mathematical model [39]. In Figure 13, the linear regression of the hydraulic heads is presented, as suggested by Anderson and Woessner [40]; it can be seen the comparison of the hydraulic head data measured in the field against the hydraulic head calculated by the model [41].

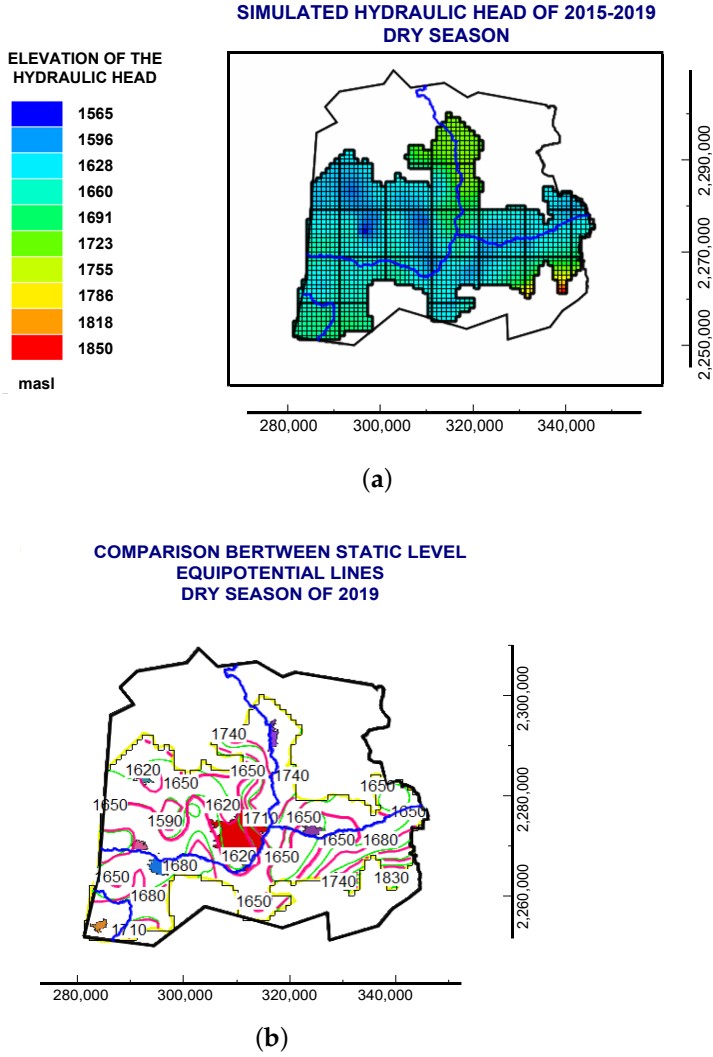

**Figure 12.** Dry season. (**a**) Values of hydraulic head in transitory state in the time interval 2015–2019. (**b**) Comparison between static level elevation equipotential lines calculated by the model (pink color) and those measured in the field (green color).

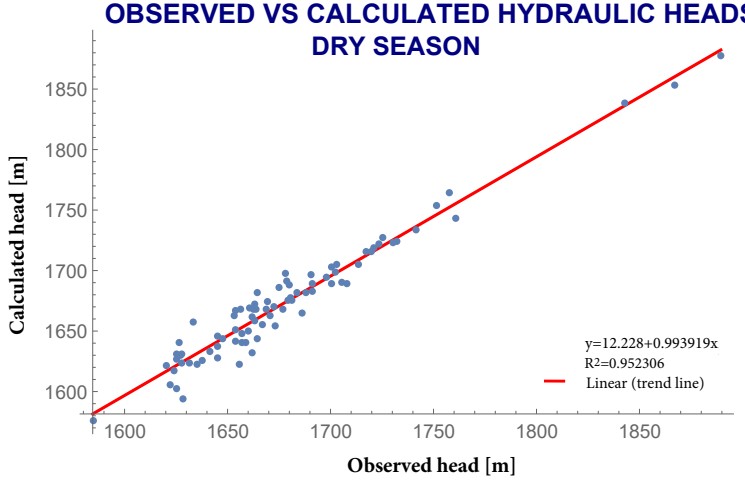

**Figure 13.** Correlation of observed and calculated piezometric levels in the dry season.

Rainy Season

For this season, the configuration of the hydraulic heads of the year 2010 was established as the initial head in order to reproduce the configuration of the year 2015 through a simulation period of 1826 days, equivalent to 5 years. The rest of the hydraulic parameters used in the simulation were the same as those of the dry seasons, except for the conductance values of the river (912.5 m²/d) and the channel (15.3 m²/d). Calibration was carried out by the trial-and-error method. Figure 14a shows the hydraulic head obtained by the model from 2010 to 2015; on the other hand, Figure 14b shows the head in terms of isolines compared to the static levels measured in the field in 2015.

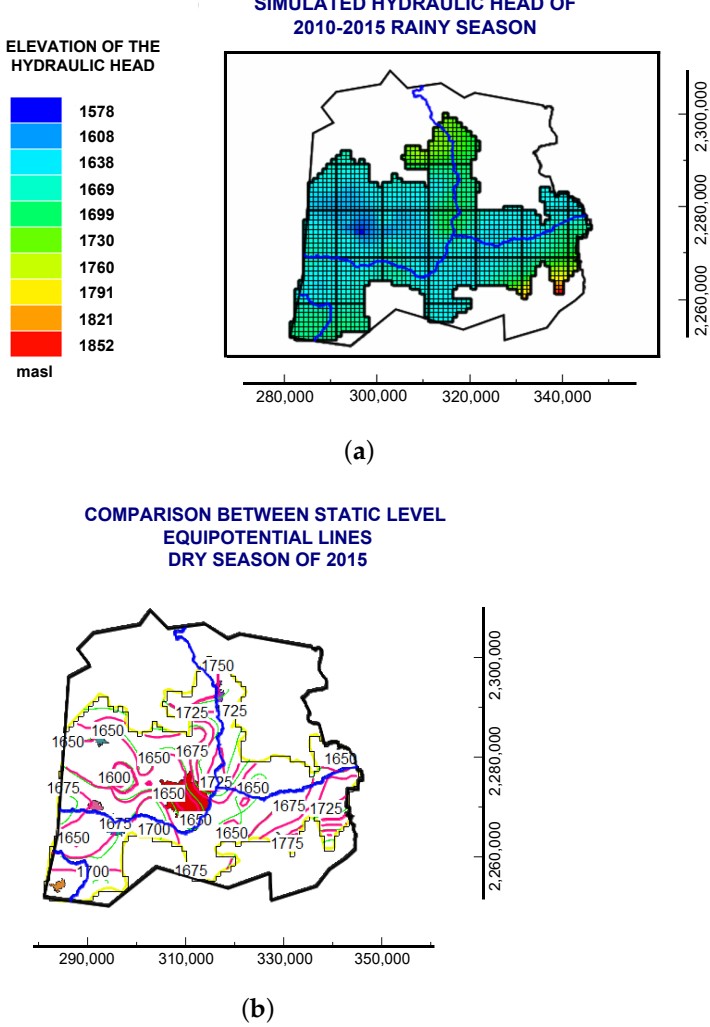

**Figure 14.** Dry season. (**a**) Values of hydraulic head in transitory state in the time interval 2010–2015. (**b**) Comparison between static level elevation equipotential lines calculated by the model (pink color) and those measured in the field (green color).

Also in this season, the linear regression of the hydraulic heads in Figure 15 shows the comparison of the hydraulic head data measured in the field against the hydraulic head calculated by the model during the 2015 rainy season.

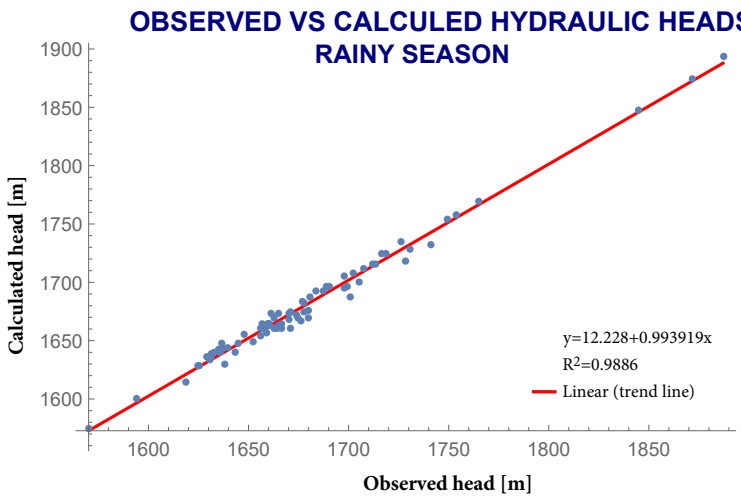

**Figure 15.** Correlation of observed and calculated piezometric levels in the rainy season.

### 2.3.3. Validation of Results

For the evaluation of the calibration results, a qualitative and quantitative analysis is made; however, it is never known if the fit between the model and reality is good enough, given that, to date, there is no standard protocol to evaluate the calibration process. There are different criteria for evaluating trial-and-error calibration, such as the comparison between contour maps of measured and simulated heads, scatter plots of measured versus simulated heads, lists including measured and simulated heads along with their differences, accompanied by some kind of average of these differences, and so on. The three most common ways of expressing the difference between measured heads and simulated heads are the mean error ME, the mean absolute error MAE, and the root mean square error RMS [42].

However, some authors, such as the creators of Waterloo Hydrogeological [43], consider that the normalized RMS is a more representative measure of the adjustment than the standard RMS since it represents the scale of the potential range of the data and is expressed as a percentage. A normalized RMS error of less than 10% indicates an acceptable calibration of the model [44]. For the case study of the dry season, a normalized RMS of 3.93% was obtained and a normalized RMS of 1.84% during the rainy season.

### 2.3.4. Sensitivity Analysis

Sensitivity analyses are conducted with the purpose of quantifying the uncertainty of the estimated parameters [3,40], time periods and boundary conditions in the calibrated model [45]. During the sensitivity analysis, the calibrated values of hydraulic conductivity, recharge, storage coefficient and specific yield were changed; in the same way, sensitivity analysis was performed for the conductance values of the inputs, the river and the channel. However, the specific performance together with the conductances of the river, canal and inputs produced very small variations for which it was decided not to present them in the graph. The ranges established to perform the analysis were ±30%, ±20% and ±10%.

For this analysis, a graph was used, where each parameter that was varied represents a series, and RMS was compared; at the origin, RMS is zero because it is calibrated. The parameters that present the greatest variation are the specific yield and the hydraulic conductivity. The values of these parameters tend to change significantly; on the other hand, the specific storage coefficient has the least variable value.

The specific storage coefficient parameter has variations of 0.028% to 0.015% with respect to the interval of +30% to −30%, so it is considered that the variation range does not represent even 1% of the hydraulic head variation, so this parameter is also insensitive to the model. On the other hand, the recharge presents a variation in the hydraulic head, which reaches up to 0.9%; therefore, this parameter does represent important changes in

the study area, but it is less than that of hydraulic conductivity (about 1.1%) and specific yield (about 1.3%). Therefore, these last two parameters can modify the hydraulic heads of the model, as they are very sensitive.

## 3. Results

The hydrodynamic simulation model that is exposed in this work represents the underground behavior of the Celaya Valley aquifer. The model operates in a transitory state covering a simulation period of 15 years for the dry season, from 2015 to 2030, while for the rainy season, it is 20 years, from 2010 to 2030, and calibrated with the information available for the years 2019 and 2015, respectively.

The calibrated model in a transient state is a good tool to improve aquifer management since it is capable of reproducing the real behavior of the system; therefore, from this model, we can determine the possible response of the system to events that alter the aquifer operation. In addition to the above, three scenarios were proposed to be simulated: trend, pumping increase and pumping reduction.

Two zones were established in order to describe in more detail the hydrodynamic behavior of the aquifer. These zones are the areas that present the greatest drawdowns; therefore, their analysis is important. Zone A is located between the municipalities of Juventino Rosas, Villagrán and Celaya; in addition, 8 control wells were used for its analysis. These are IGC-1004 (1), IGC-1298 (2), IGC-1768 (3) , IGC-726 (4), IGC-766 (5), IGC-769 (6), IGC-787-A (7), and IGC-789 (8). Zone B is located between Celaya, Comonfort and Apaseo el Alto, and the wells used for the analysis were IGC-1084 (1), IGC-541 (2), L-1119 (3), L-159 (4), L -183 (5), L-184 (6), L-597 (7) and L-766 (8); this applies to each scenario presented.

### 3.1. Trend

In this scenario, the model was simulated without any type of modification, that is, the current rate of water extraction continues until the year 2030. Thus, no modification is exerted on the hydrological components in order to estimate and observe the behavior of the piezometric levels.

### 3.1.1. Dry Season

Figure 16 shows the configuration of the hydraulic head obtained for the trend scenario. It is observed that the piezometric levels decrease for the year 2030, at a rate of meters. In Zone A, where CONAGUA and CEAG have reported subsidence of the land due to the massive extraction of groundwater, it can be seen that the depletion cone has become more pronounced, that is, there is a decrease in the piezometric level and the cone has elongated in a northeasterly direction; compared to 2019, it falls from 3.3 m to 40 m. On the other hand, in Zone B, considerable drawdowns ranging from 20 m to 40 m are also observed.

In Figure 17a, some control points that were selected to recognize the behavior of the hydraulic heads of Zone A are observed. The red line corresponds to the variations of the hydraulic head of the period 2019–2030, that is, calculated difference in heads of the simulation of the year 2019 with that of 2030, with the exception of two wells: IGC-1046 and IGC-1298. The hydraulic heads decrease as time goes by, which can be associated with the fact that the control points used are representative of the areas where there are large abatements. The blue line represents the head difference of the year 2015 and 2019, and in this period, it is still possible to see both positive and negative head variations.

In Figure 17b, the control points for Zone B are displayed; the behavior of the wells in the period 2019–2030 is similar to that of Zone A, that is, these wells lose head with the passage of the years, while for the 2015–2019 period, only four wells were in decline, and the rest recovered.

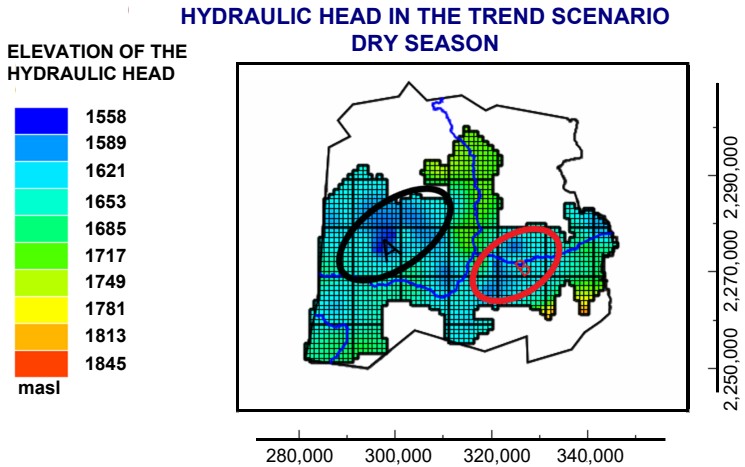

**Figure 16.** Elevation of the hydraulic head of the year 2030 through the trend scenario in the dry season.

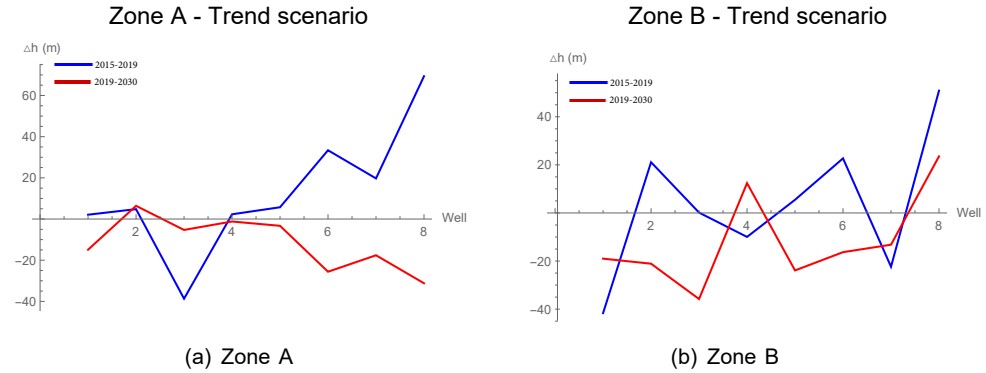

**Figure 17.** Variation of the hydraulic head in the trend scenario of the dry season.

In general terms, the equipotential lines continue to maintain similarities with respect to the configuration of the hydraulic head of 2019, simply the cones that were present this year have been longer and their depths have increased, mainly in Zone A and B, the rest of the aquifer has a similar behavior to the year 2019 (Figure 18).

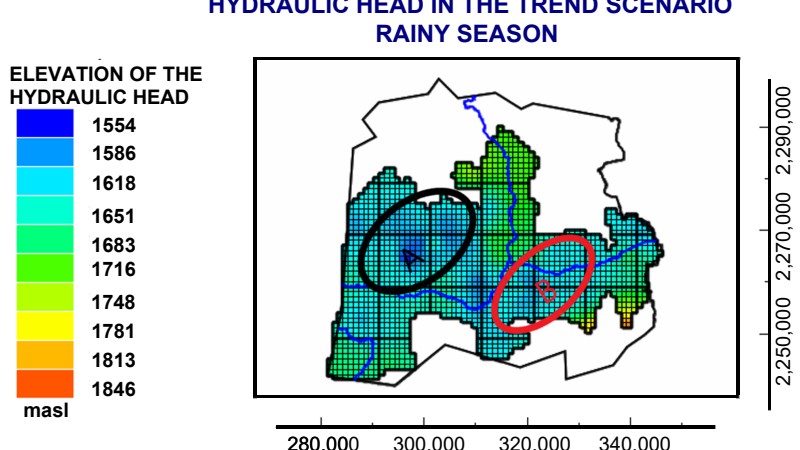

**Figure 18.** Elevation of the hydraulic head of the year 2030 through the trend scenario in the rainy season.

### 3.1.2. Rainy Season

The configuration of the hydraulic head projected for the year 2030 can be seen in Figure 19; it is notorious that, when making the comparison with the dry season, the piezometric levels increase by a few meters. In Zone A, the depletion cone can still be seen. Presumably, although there is more recharge in the aquifer, it is not enough to achieve recovery in the entire aquifer; however, in Zone B, it is no longer so visible the formation of dejection cones. The rainy season especially benefits this area.

### 3.2. Pumping Increase

Currently, there is a high population demand in the urban area, mainly in Celaya and Juventino Rosas, which indicates a greater demand for water resources, mainly underground; therefore, in this scenario, a 25% increase in pumping extractions was simulated in all the sectors that extract water from this aquifer. This scenario simulates an unfavorable situation in the region.

### 3.2.1. Dry Season

It can be seen in Figure 19 the configuration of the hydraulic head obtained for scenario 2, corresponding to the increase in pumping to 25%. This applies to any type of use that is given to the extractions (agricultural, urban public, industrial, others).

Zone A becomes extremely vulnerable when pumping increases. It can be seen that the depletion cone has extended toward Celaya and therefore, there is a decrease in the hydraulic head in the rest of the aquifer. The decrease goes from 20 m to 60 m. Zone B does not show any recovery of the hydraulic head with respect to the year 2019; the decrease in the hydraulic head reaches 70 m.

It is also possible to see a new area, called Zone C, located northeast of Comonfort, right on the border with the state of Querétaro, where it is possible to see the heads decrease; therefore, we see the formation of a new depletion cone that was not seen in the trend projection.

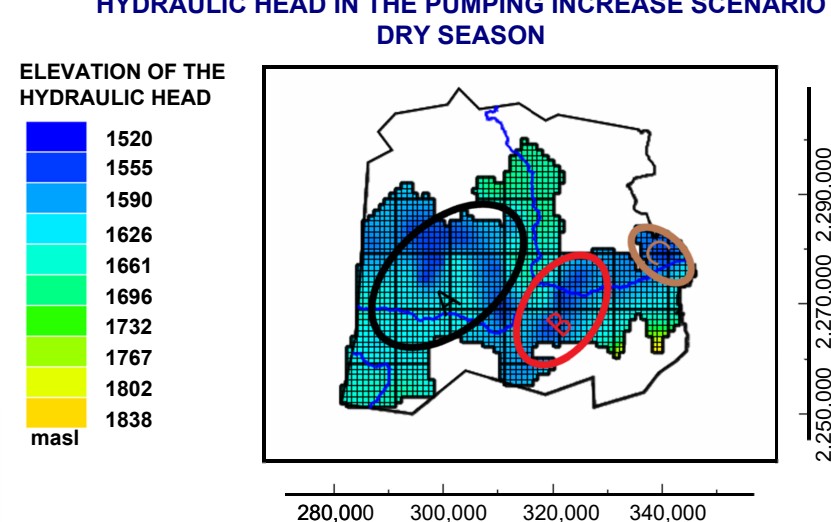

**Figure 19.** Elevation of the hydraulic head of the year 2030 through the pumping increase scenario in the dry season.

In Figure 20a, we can visualize the behavior of the control points of Zone A. When comparing the 2015–2019 period with 2019–2030, it is clear to see recovery in the first period, while in the second period in this area, there are only reductions that exceed 50 m in just 11 years. Figure 20b shows the control points for Zone B; the behavior of the wells in the 2019–2030 period is similar to that of Zone A, there is no recovery, and the losses amount to 70 m. In addition to Zones A and B, a new zone can be recognized in which

a depletion cone begins to form at the northeastern limit of the aquifer, which reaches a height of 1560 m above sea level.

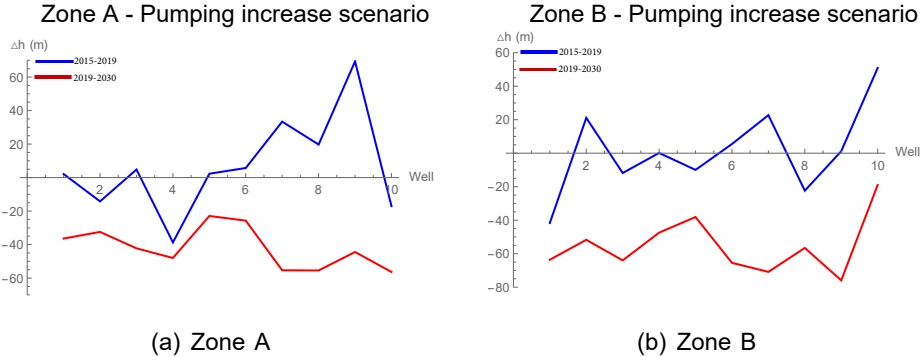

(a) Zone A  (b) Zone B

**Figure 20.** Variation of the hydraulic head in the pumping increase scenario of the dry season.

3.2.2. Rainy Season

The projection of the hydraulic head for the year 2030 in the pumping increase scenario can be seen in Figures 20 and 21. Making the comparison between rainy and dry seasons, it is established that the behavior of the hydraulic head between both seasons varies from 4 to 20 m, but also in this season, there are still considerable drawdowns. For example, in Zone A, the cone of abatement is elongated from west to east; in Zone B, the abatement extends from the southwest of Celaya to the north of Comonfort; and in Zone C, a new depletion cone is formed.

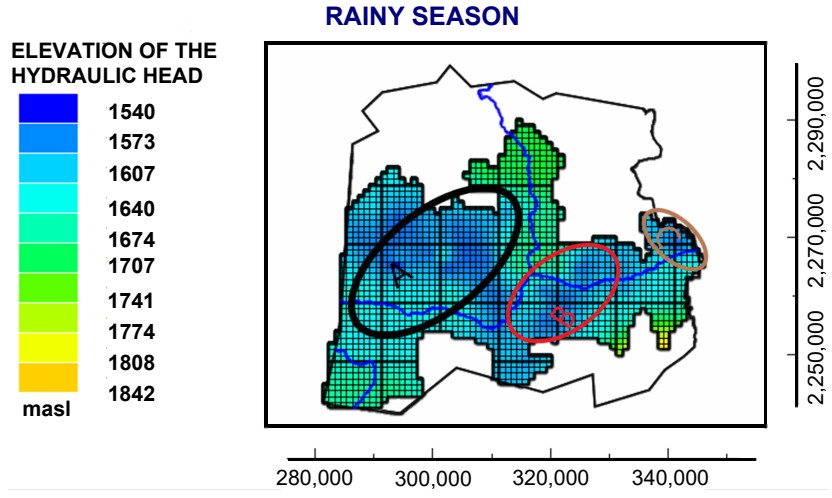

**Figure 21.** Elevation of the hydraulic head of the year 2030 through the pumping increase scenario in the rainy season.

*3.3. Pumping Reduction*

This aquifer has a deficit in the availability of water, and it is known that the CEAG is taking measures to control this problem and subsequently provide a solution. Therefore, we set out to simulate a 50% reduction in pumping for all the sectors that extract: agricultural, public–urban, industrial and others. The purpose of this scenario is to minimize the effect of massive pumping extractions on the aquifer and simulate a favorable situation for the region.

### 3.3.1. Dry Season

As CONAGUA has documented, the main source of extraction from this aquifer is through pumping, and this scenario has shown that a large part of the recovery of this aquifer depends on the reduction of extractions for various activities.

In Figure 22, it can be seen that the cone located in Zone A is still present; however, it has decreased its extension and is no longer as noticeable as in 2019. In Zone B, we can see that the behavior of the cone has changed; it is no longer so pronounced, which is due to the general recovery of the aquifer. Between Jaral del Progreso and Cortázar, a small cone can be seen, Figure 22, delimited by an elevation of 1670 meters above sea level. It is possible that, with the tendency to reduce pumping, this zone of the aquifer could recover.

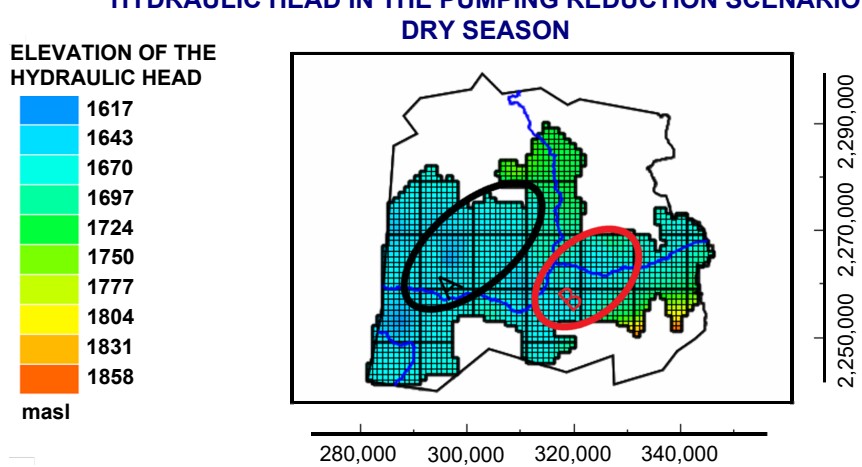

**Figure 22.** Elevation of the hydraulic head of the year 2030 through the pumping reduction scenario in the dry season.

In Figure 22, we can see a favorable situation in the behavior of the control points of Zone A, the zone where there are currently depletions. With this trend, they tend to recover, and the maximum recovery of the water levels reaches 90 m, while the minimum recovery is 25 m. This is a good result for the rest of the aquifer in the second period. Also shown recovery at the control points for Zone B, the maximum and minimum recoveries of this zone are 80 m and 20 m, respectively (this increase can be seen in Figure 23).

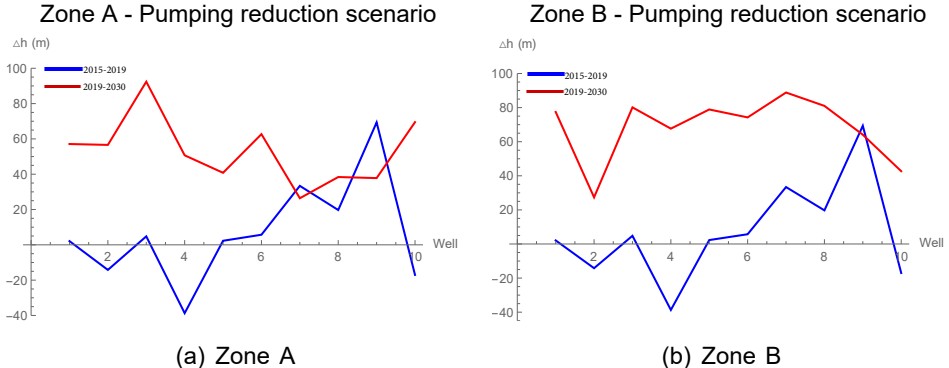

**Figure 23.** Variation of the hydraulic head in the pumping reduction scenario of the dry season.

### 3.3.2. Rainy Season

As described in the dry season, the recovery of the aquifer in this projection is remarkably visible, see Figure 24. In Zone A, there is still the depletion cone; however, the elevation of the hydraulic head has increased to 1650 masl, and with this trend, a partial recovery of the area is possible in the coming years. In Zone B, we can visualize the drawdown

reduction; therefore, the direction of the flow follows its trajectory to the Valley. Recharge zones in the aquifer continue to be marked north of Apaseo el Grande and east of Apaseo el Alto.

**Figure 24.** Elevation of the hydraulic head of the year 2030 through the pumping reduction scenario in the rainy season.

*3.4. General Analysis of Simulation Scenarios*

Through the different simulated scenarios, it has been determined that the evolution of the hydraulic head shows decreases, mainly in zones A and B, that is, part of the problem related to the decreases in the static level is due to the massive extractions by pumping water. The trend projection for the year 2030 in the aquifer is worrying, given that two areas highly vulnerable to depletion have been identified, where the minimum elevation reached by the hydraulic head is 1579 m above sea level in the dry season and 1580 m above sea level in the rainy season; this indicates that year after year, it will continue to decline, and exploitation of the aquifer will be non-sustainable in the long term.

Zones A and B are defined as the more vulnerable areas in the aquifer, while in the east of the aquifer, there is more stability of the hydraulic head, regardless of the projection used; however, the possibility of making a new evaluation of the assignment of the hydrogeological parameters used in this model could be made in order to corroborate the results obtained.

The simulations of the dry and rainy seasons have served to make a comparison of the behavior of the hydraulic head when there is a greater amount of water entering the aquifer and how the flow lines behave in each situation. With the projections, it has been identified that the variation of the static level between both seasons is, on average, 5 m, which means that the aquifer, in some months (June, July, and August) presents greater head, and in the driest months (March, April, and May), there is a greater decrease in the hydraulic head.

**4. Conclusions**

We studied the Celaya Valley aquifer; through the piezometric history provided by the CEAG, we were able to develop a numerical model of the hydraulic potential and its time evolution in this aquifer. Once calibrated, by statistic comparison of the heads measured in the field and the ones estimated by the model as explained before, the model allowed us to analyze the behavior of the system. Qualitatively, it is possible to determine the existence of direction changes in the equipotential lines between each simulation period. The components of the hydrogeological balance were calculated in this work for the dry season as well as for the rainy season, which resulted in a deficit of 170 mm$^3$/year and 118.3 mm$^3$/year, respectively, indicating that the outputs are greater than the inputs and that the aquifer presents serious problems of overexploitation. The elevation of the

minimum hydraulic head reached in the 2015 dry season was 1572 m above sea level, while in the rainy season it was 1585 m, differing by 13 m between seasons of the same year. Regarding the recharge zones, the simulations allowed us to identify two inputs, one to the north of Apaseo el Grande and another to the southeast; these inputs are present in both seasons. In the rainy season, the abatement of the area located between Celaya, Juventino Rosas and Villagrán is clearly distinguished, while during the dry season, in addition to this area, a cone of abatement becomes apparent to the south of Celaya; it is defined by a 1590 m above sea level hydraulic head, an effect that cannot be seen in the rainy season. This last argument reinforces the idea of the aquifer's overexploitation. We must also state that the piezometric levels are highly variable since 2013; therefore, its trend is unstable over time. Nevertheless, we could reproduce in the simulation model the hydraulic behavior of the piezometric levels up to 93% accuracy in both seasons, and thus we were able to carry out different simulation scenarios. Running the simulations up to 2030, it can be seen throughout both seasons the presence of two zones that are highly vulnerable to massive water withdrawals from the aquifer. These were identified as Zones A and B. Heads in the dry season for Zone A drop approximately 10 m more than in the rainy season. Zone B also has head drops from 10 to 15 m between both seasons, being more favorable in the rainy season. The zones with the most recharge potential are still the same as in 2015. From the proposed scenarios that were studied, we are able to conclude that there are two zones where the piezometric levels are unstable due to large pumping extractions, it being necessary to drastically reduce the number of pumping wells at least 50%. This percentage is very high, but with this reduction, it is possible to observe a recovery in the system, mainly in Zones A and B; a lower percentage of pumping reduction does not improve the deficit in the aquifer. We must take into account that pumping is not the only cause of hydraulic head reductions in the aquifer, but it considerably improves its depletion. Upon performing a sensitivity analysis, we found that the model is highly sensitive to three main components, hydraulic conductivity, specific performance and recharge. The rest of the parameters, such as the conductances of the river and canal, did not present significant variations. Finally, we regard three main lines of future work. It would be interesting to consider structural geological features, such as faults and fractures, in order to improve the approximation of the model to the real geometry and geology of the aquifer; another possibility consists in the incorporation of local values for the geological parameters, i.e., we enrich the model by taking into account the presence of certain inhomogenieties within the geological cross cut that we used; and, at last, it is desirable to feed the model with the values of some hydrogeochemestry parameters in order to make a more profound study of the aquifer and the water itself.

**Author Contributions:** Conceptualization, A.B.R.-A., J.A.R.-L. and V.M.V.-B.; Investigation, A.B.R.-A., J.A.R.-L., V.M.V.-B. and J.I.R.M.; Writing—original draft, A.B.R.-A., J.A.R.-L., V.M.V.-B. and J.I.R.M.; Writing—review & editing, A.B.R.-A., J.A.R.-L., V.M.V.-B. and J.I.R.M. All authors have read and agreed to the published version of the manuscript.

**Funding:** This research was funded by VIEP-BUAP and CONACyT, A.B. Rubio-Arellano thanks CONACyT for the studies grant for this work.

**Data Availability Statement:** Not applicable.

**Acknowledgments:** The author thank CONAGUA and CEAG for their support.

**Conflicts of Interest:** The authors declare no conflict of interest.

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
