# Peer review of "Modeling the Groundwater Dynamics of the Celaya Valley Aquifer"

_water, doi:10.3390/w15010001_

Round 1
Reviewer 1 Report
The research is interesting and dounds well with the objectives set are professinally realized.
Author Response
We thank the comments kindly provided by the reviewer.
Reviewer 2 Report
Line # |
Comment |
42 |
‘decade-fashion’ , cannot seem to understand what you mean to say |
Fig 2 |
Geology map of the VP aquifer (till now we only knew of CV aquifer, where does VP come from?) |
114 |
‘As is know’, change ‘know’ to ‘known’ |
247 |
‘cells’ or ‘wells’ ? |
258 |
‘registration period’ requires elaboration |
Fig 10 |
presenting model output in a research paper is inappropriate as space in a journal is scarce and we should use it judiciously. Plz remove this figure. You have described the steady state simulation characteristics in detail in text, so no need for a figure. |
Fig 13(a) |
kindly remove the mesh grid so that underlying distribution of heads can be seen clearly |
451-466 |
the discussion can be easily made brief which your article requires very much. You present standard RMS error measure but then quote Waterloo Hydrogeological [35] to weaken its credibility. I suggest that you should remove reference to standard RMS error and straightaway present results of normalized RMS error as it is a more suitable measure. |
488 |
add ‘specific’ before ‘yield’ |
|
Fig 17 caption uses the word ‘analysis’ but on line 468 ‘analyzes’. Change later to ‘analysis’ |
Fig 17 |
no label on x-axis |
Fig 17 |
only shows that when you vary heads by ±30% there is a max change of about 1.3% in specific yield, 1.1% in hydraulic conductivity and about 0.9% in recharge. You can provide this information in text. It does not justify inclusion of a figure. |
Fig 16 |
no x and y label provided |
|
In the line above Fig 16 caption ‘h!]’ appears. What purpose does it serve? |
520 |
‘at a rate of meters’. One should expect some figure between ‘at a rate of’, and, ‘meters’? |
Author Response
Comment about line 42: What we mean to say is that field studies are performed about every ten years. So, we have rewritten the statement as “This is done since actual monitoring of the state of the aquifer is expensive and performed (when done) through field studies around every ten years in Mexico, as in other developing countries.”
Comment about Fig. 2: This is indeed a typographical error, we have corrected it.
Comment about line 114: We have made the kindly requested change.
Comment about line 247: The correct term is “cells” since we are talking about the horizontal inputs, we determined this number of cell as those corresponding to high elevation zones and with the aid of the isolines of the piezometric level. We have made the adequate change.
Comment about line 258: What we mean to say here is that the stations have acquired data during different time periods, since this is due to the moment each station began to work. We have made the suitable elaboration.
Comment about line Fig 10: We agree with the point made in the comment, so we have removed this figure.
Comment about Fig 13(a): We would gladly remove the mesh grid, nevertheless this would imply that we rerun the simulations in order to obtain new graphics, thus we are unable to make the recommended change. We apologize.
Comment about line 451-466: We appreciate the suggestion and have made the adequate change.
Comment about line 488: We have made the suggested addition.
Comment about Fig 17: We have made de adequation by removing the figure as suggested and mentioning the results in the text.
Comment about Fig 16: We have made the corrections to the axes and removed the “h!]” characters since they are a typographical error.
Comment about line 520: Indeed, there was a figure missing we have made the correction.
Reviewer 3 Report
The abstract should be revised as follows:
"We propose a hydrodynamic model to understand the piezometric operation of the Celaya Valley aquifer. The aquifer is located east of Guanajuato State in México. Our proposed model reproduced the aquifer's performance under transitory conditions during the dry season from 2015 to 2019 and the rainy season from 2010 to 2015. The simulation was projected for the two seasons up to eleven years ahead and under three different simulation scenarios: trend, pump reduction, and pump increase. In general terms, the model accurately reproduces the natural conditions of the aquifer, and it is necessary to continue taking measures for the preservation of water; similarly, it is suggested to continue monitoring the piezometric aquifer levels to update the model based on data availability."
The authors claimed that "The simulation was projected for the two seasons up to eleven years ahead and under three different simulation scenarios." I suggest that authors explain how they can project for the two seasons up to eleven years only if under three different simulation scenarios.
Author Response
We apologize if our writing in the abstract was confusing since the simulations can be made under any scenario one choses, we consider only three of them, however we have rewritten this part of the abstract. We expect it reduces the chance of misunderstanding what we want to mean.
Round 2
Reviewer 2 Report
The authors have incorporated my suggestions & corrections in the revised text. I am satisfied with the article's quality.
Reviewer 3 Report
The authors have well addressed my comments. I suggest this manuscript can be accepted for publication in the Water journal.